# Bio-Inspired Nanocarriers Derived from Stem Cells and Their Extracellular Vesicles for Targeted Drug Delivery

**DOI:** 10.3390/pharmaceutics15072011

**Published:** 2023-07-24

**Authors:** Munire Abudurexiti, Yue Zhao, Xiaoling Wang, Lu Han, Tianqing Liu, Chengwei Wang, Zhixiang Yuan

**Affiliations:** 1College of Pharmacy, Southwest Minzu University, Chendu 610041, China; m13579500259@163.com (M.A.); wxl3232@sina.com (X.W.); lunahan@swun.edu.cn (L.H.); 2Department of Pharmacy, Sichuan Tianfu New Area People’s Hospital, Chengdu 610213, China; zytfxq2018@163.com; 3NICM Health Research Institute, Western Sydney University, Westmead 2145, Australia; michelle.tianqing.liu@gmail.com; 4Division of Internal Medicine, Institute of Integrated Traditional Chinese and Western Medicine, West China Hospital, Sichuan University, Chengdu 610041, China

**Keywords:** bio-inspired, stem cell, extracellular vesicles, targeted drug delivery, nanocarriers

## Abstract

With their seemingly limitless capacity for self-improvement, stem cells have a wide range of potential uses in the medical field. Stem-cell-secreted extracellular vesicles (EVs), as paracrine components of stem cells, are natural nanoscale particles that transport a variety of biological molecules and facilitate cell-to-cell communication which have been also widely used for targeted drug delivery. These nanocarriers exhibit inherent advantages, such as strong cell or tissue targeting and low immunogenicity, which synthetic nanocarriers lack. However, despite the tremendous therapeutic potential of stem cells and EVs, their further clinical application is still limited by low yield and a lack of standardized isolation and purification protocols. In recent years, inspired by the concept of biomimetics, a new approach to biomimetic nanocarriers for drug delivery has been developed through combining nanotechnology and bioengineering. This article reviews the application of biomimetic nanocarriers derived from stem cells and their EVs in targeted drug delivery and discusses their advantages and challenges in order to stimulate future research.

## 1. Introduction

Stem cells are undifferentiated cells in the human body that can differentiate into various functional cells [1]. Because stem cells can undergo self-renewal through cell division, this sets stem cells apart from other types of cells [2]. The ability to self-renew, the capacity to divide and produce differentiated offspring, the potential for differentiation, and the capability to replicate cells with the same potency as themselves are also known as the essential characteristics of stem cells [3]. These properties allow stem cells to treat diseases for clinical application [4]. However, the majority of the information on stem cells used in the treatment of diseases comes from small randomized trials, and does not show satisfactory therapeutic outcomes. In addition, stem-cell-based therapy may increase the risk of immune rejection and malignant transformation in clinical treatment [5]. Due to these factors, there are still many limitations to be overcome and the expected therapeutic level has not been fully achieved [6].

Extracellular vesicles (EVs) secreted by stem cells are crucial mediators to regulate the effect of stem cell therapy [7]. They are small nanoparticles composed of various complex molecules, including proteins, nucleic acids, and lipids, which can be delivered to recipient cells [8,9]. They have recently been considered ideal drug carriers in recent years due to these characteristics and have been widely used in targeted drug delivery applications [10]. In-depth research is being carried out for the development and improvement of EV-based targeted drug delivery systems.

Targeted drug delivery is a technique used to deliver drugs to specific tissues or organs of patients in systemic administration, while reducing their accumulation in healthy tissues [11]. This leads to precise and efficient drug delivery with fewer side effects after local or systematic administration. In vivo drug delivery has greatly benefited from the advancement of nanotechnology [12]. However, traditional synthetic nanocarriers may have a number of drawbacks, such as low degradability, low biocompatibility, and potential toxicity, that prevent them from fully meeting the needs of disease treatment [13]. It becomes a formidable challenge and hot spot to develop effective, safe, and accurate drug delivery to overcome these shortcomings. Stem cell membranes and EVs have been extensively studied and applied as nanocarriers for drug delivery. These cell-derived nanocarriers have excellent biocompatibility, good stability, low immunogenicity, and inherent targeting or tendency ability (e.g., penetration of biological barriers such as the blood–brain barrier). Moreover, while maintaining the function of stem cells, they have long circulation time in vivo due to less phagocytosis, suggesting stem-cell-derived EVs are outstanding candidates for targeted drug delivery and disease treatment [14]. However, their intrinsic drawbacks, including low extraction yields, poor recovery, purity, and encapsulation efficiencies, restricted the large-scale production of therapeutic exosomes in the clinic [15]. Therefore, bionic EV mimetic nanovesicles (MNVs) have garnered attention. They contain key components of natural EVs and exhibit similar properties, with higher production yield compared to EVs [16]. In this review, we investigate the applications of biomimetic nanocarriers derived from stem cells and their EVs for targeted drug delivery and discuss their advantages and challenges in order to stimulate future research.

## 2. Characteristics and Function of Stem Cells

Stem cells are undifferentiated cells with strong differentiation potential [17,18]. They can differentiate into various special cell types and, thus, form different tissues or organs under certain environments and carry out a series of self-renewals through this division ability, which sets them apart from other cell types [19]. According to differentiation ability, stem cells can be divided into totipotent stem cells, pluripotent stem cells, multipotent stem cells, oligopotent stem cells, and unipotent stem cells [20].

Stem cells are used as repair systems in the body, as they have a remarkable capacity to induce de novo tissue formation and promote the repair and regeneration of organs in the body [21]. They provide therapeutic effects through regenerating damaged cells to help organ recovery and improve body development. Based on the natural capabilities of stem cells, researchers have utilized their biological mechanisms to carry out stem-cell-based therapies [22]. They are actively used to improve tissue regeneration, particularly for hematological diseases and skin regeneration [23,24]. For example, mesenchymal stem cells (MSCs) show positive effects in the treatment of burn skin due to their multi-lineage differentiation characteristics. MSCs play a healing role through secreting mediators (e.g., biologically active factors) that can reduce skin inflammation when repairing burnt skin. In addition, stem cells have shown great potential in treating neurodegenerative diseases such as Alzheimer’s disease (AD), skin diseases such as vitiligo, musculoskeletal diseases such as osteoarthritis, congenital cardiovascular diseases and blood cell diseases such as leukemia, as well as other diseases such as diabetes [25,26,27].

However, stem cells have the disadvantages of possible immune rejection, low recovery rate, and high contamination to viruses, which result in inevitable treatment flaws and risks [6,28]. In order to overcome these issues, the use of nanotechnology is growing in current research, and stem-cell-related nanotechnology has also received extensive attention. Using nanotechnology to modify the membrane of stem cells can significantly enhance their targeting delivery and achieve a better therapeutic outcome for treatment [29,30].

## 3. Bio-Inspired Nanocarriers from Stem Cells

It is crucial to investigate effective strategies for drug administration, targeted accumulation to diseased organs, and controlling the rate of drug release. Nanomaterial-based drug delivery offers solution for advanced drug packaging and precise tissue accumulation [31]. Through the engineering of the physical and chemical properties as well as surface ligands of engineered nanoparticles, they can penetrate biological barriers and enhance drug delivery efficiency [32]. However, nanotechnology as a drug delivery approach still has limitations [33]. Magnetic nanoparticles, polymer nanogels, and other non-natural drug carriers are often categorized as “non-self” and are rapidly cleared by the immune system. They lack the ability to actively sense the disease environment, leading to an insufficient accumulation of drugs in the desired organs or tissues, resulting in low targeting efficiency. Additionally, they present challenges such as immunogenicity and toxicity [34]. Therefore, there is a high demand for more precise nanomedical technologies to overcome these issues [35]. Stem cells, as natural carriers, can circumvent the issues associated with “non-natural” carriers.

Stem cells exhibit inflammation-driven tumor tropism mediated by adhesive ligands such as Sialyl Lewis X (SLeX) and P-selectin glycoprotein ligand-1 (PSGL-1). The presence of surface antigens and innate targeting ligands greatly favors their use as tumor-targeted carriers. Currently, stem cells have been employed for the delivery of protein and peptide-based anticancer drugs, such as interferons and interleukins. Stem cell nanocarriers are highly favored for their safety profile in terms of non-oncogenicity [36]. In recent years, the stem cell membrane has been used as a natural coating material, taking advantages of core nanoparticles and source cells in order to achieve highly efficient targeted drug delivery [37,38]. Due to the abundant expression of chemokines and chemokine receptors on stem cell membranes, they can interact with target cells, such as those at the site of inflammation. Therefore, this type of nanocarrier can further enhance the potential for effective drug delivery in conditions such as inflammation [39].

### 3.1. Cell Membrane Modification

The modification of the cell membrane surface has been proven to improve treatment delivery efficiency [40]. Furthermore, stem cell membranes possess an inflammatory homing effect, which enhances their ability to target and accumulate in inflamed tissues. This effect can be leveraged to further enhance the therapeutic potential of nanomedicines [41]. Generally, there are three methods available for modifying the nanoparticle membranes of stem cells to promote therapeutic applications of biomimetic nanomedicines, including physical, chemical, and biological modifications. Due to the membrane fluidity of cell membranes, materials such as glycosylphosphatidylinositol can be easily anchored on the membranes as membrane coating. However, this physical modification method also has disadvantages, such as low stability and insufficient coating efficiency (Figure 1A) [42]. Cell membranes are rich in proteins with thiol and amine residues and polysaccharides with hydroxyl residues, which can serve as sites for different covalent coupling reactions [43]. Compared with the physical modification method, this strategy is more direct, stable, and efficient, with a reduced risk of altering membrane permeability and functions (Figure 1B). The biological modification approach utilizes genetic tools to generate the desired proteins or peptides on the stem cell membranes, which then form coated nanocarriers [44]. Compared with the first two methods, this biological modification provides more modification options, allowing for both up-regulation and down-regulation of target proteins on the cell membrane (Figure 1C) [45]. Through modifying stem cell membranes, it is possible to create membrane-coated nanoparticles with specialized functionalities that go beyond what the cell membrane alone can provide [46]. The modified membrane allows the encapsulated nanoparticles to possess the complex and distinctive surface physicochemical properties of stem cells, thereby extending their blood circulation time, enhancing active targeting, and improving cellular internalization. This nanocarrier system can address challenges associated with biocompatibility, immune responses, and off-target effects caused by nanoparticles [47].

### 3.2. Cell-Membrane-Coated Nanoparticles

Over the years, cell membranes have been extensively utilized in the fabrication of nanocarriers [48], and cell-membrane-coated nanoparticles have emerged as delivery vehicles [49]. Cell-membrane-coated nanoparticles have significantly improved the targeted delivery performance of cell membranes, reduced the immunogenicity of the cell membrane component, and overcome the limitations of traditional nanotechnology [47]. Membrane-coated nanoparticles can not only replicate the highly complex functions necessary for stem cells to achieve effective biological interfaces but also maintain the small size and loading characteristics of nanoparticles [50,51]. The coating strategies of cell membrane nanoparticles include the separation of membrane vesicles and the fusion of membrane vesicles with core nanoparticles [52]. Cell-membrane-based targeted delivery systems have emerged as promising strategies due to their advantages [53]. Various types of stem cell membranes, such as MSCs, have been reported as effective natural carriers. Cellular-membrane-camouflaged biomimetic nanoparticles have become an effective targeted drug delivery approach due to their natural structural advantages [54].

### 3.3. Bio-Inspired Nanocarriers

#### 3.3.1. MSCs

MSCs are a type of multipotent stem cell that possess functions such as immune regulation and the release of growth factors [55,56]. Owing to their self-proliferation and self-renewal, they are easy to obtain and can easily grow in culture dishes [57]. Their unique advantages include excellent low immunogenicity [58], the ability for multidirectional differentiation [59], and migration and homing, which contribute to their significant clinical potential. Consequently, they have emerged as a highly promising and appealing treatment strategy for various diseases. As a result of these benefits, MSCs have become the most widely used in disease and injury research [60] and meanwhile played an important role in various inflammatory diseases [61], liver diseases, lung diseases [62], hematopoietic diseases [63], autoimmune diseases, and other diseases [64]. The MSC membrane contains growth factor receptors, chemokine receptors, cytokine receptors, and many other useful receptors that mediate signal transduction in the cellular environment and have an impact on cell metabolism and function [65]. MSCs or their membranes combined with bioengineering are focused on achieving improved therapeutic effects [45].

Yukiya Takayama et al. used the avidin–biotin complex method to modify the surface of MSCs with doxorubicin-loaded liposomes [66], and the delivery efficiency and antitumor efficacy of modified MSCs were systematically evaluated. The results showed that MSCs were modified to target tumors as effective anticancer drug delivery carriers that can enhance the tumor-targeted therapeutic system for the intercellular delivery of doxorubicin. Kartogenin (KGN), which can effectively inhibit cartilage degeneration, was loaded into Fe_3_O_4_ nanoparticles. Then, the researchers fused it with the natural bone marrow mesenchymal stem cell (BMSC) membrane, which was separated and processed into a capsule, and injected into the knee joint through the joint. The high-efficiency delivery of KGN molecules in the BMSC membrane while maintaining the biological activity of the BMSC membrane shows great potential for cartilage regeneration therapy [67,68]. Although studies have demonstrated that cell membranes as carriers can become a promising means of drug delivery, these are still in the initial stage [69]. Due to biosafety considerations, a large number of long-term studies are still needed in the future to achieve the clinical transformation of MSC membrane drug delivery [70]. However, owing to these unique advantages, the future of stem cell membrane drug delivery is bright. It is a promising clinical treatment strategy for nano-drugs as a new drug delivery platform for treating diseases.

#### 3.3.2. iPSCs

Induced pluripotent stem cells (iPSCs) are a kind of pluripotent stem cells. They have great potential to differentiate into any cell and are being studied to treat intractable diseases [71,72]. Nowadays, they are widely used, are the focus of some research such as cell therapy, and have greater potential for expansion and self-renewal than cultures of other stem cells. However, iPSCs are limited and hindered in regenerative medicine and other research fields because of their inherent tumorigenicity [73]. But, encouragingly, it has been found that iPSCs have the potential to be used as a regenerative medicine tool for the treatment of kidney disease. MSCs derived from iPSCs are easy to expand. The nanovesicles prepared via the iPSC-MSC method showed no toxicity or immunogenicity and were stable after storage. They were used to load the chemotherapy drug doxorubicin for use in triple-negative breast cancer (TNBC). The results showed that the cytotoxicity of doxorubicin to drug-resistant TNBC cells was improved through loading iPSC-MSCs, and the incidence of TNBC in mouse models was significantly reduced, which is expected to improve TNBC therapy [74,75]. This indicates that nanovesicles made of iPSC-MSCs are promising carriers for the targeted delivery of anticancer drugs, which enhances the potential of stem-cell-based targeted drug delivery platforms.

There are few reports on the use of other types of stem cells as biomimetic nanocarriers, but the advantages of most stem cells have been actively developed and utilized. Even so, there are several limitations. The general assumption inherent in most of these coating methods is that the cell membrane uniformly covers the entire NP surface, forming a complete core–shell structure. Unfortunately, research has found that the ratio of full coating never exceeds 20%, indicating that the vast majority of biomimetic nanoparticles are only partially coated [76]. This is because biomimetic programs may result in a lack of integrity in the reassembled cell membrane coating. In addition, coated and uncoated NPs are difficult to separate due to limitations in manufacturing methods [13,77]. Encouragingly, there has been development research on the endocytic entry mechanism of nanoparticles, providing a framework for understanding the internalization of cell-membrane-coated nanoparticles. This progress promotes the rational design of biomimetic nanosystems and paves the way for the development of more effective nano-drugs [76]. It is anticipated that in future research, stem cells will emerge as increasingly powerful and promising tools in targeted drug delivery systems.

## 4. Bio-Inspired Nanocarriers from Stem-Cell-Derived EVs

### 4.1. EVs Classification

EVs are vesicles secreted by the body and are biomimetic materials derived from cells themselves [78]. Initially regarded as mere waste disposal units for cellular debris, these vesicles have garnered increasing interest in scientific research due to their capacity for intercellular communication. Furthermore, there is more research that they can outperform stem cells for more targeted drug delivery therapies [79,80,81]. EVs are nano-sized membrane vesicles with a lipid bilayer structure secreted by cells with a diameter of 50–2000 nm [82], and can be secreted by almost all cell types [83]. After secretion, it will be released into the extracellular environment and then participate in the transport of cellular goods and information transmission [10,84]. EVs are classified into three types according to biogenesis and size, which are exosomes (40–100 nm), microvesicles (100–1000 nm), and apoptotic bodies (500 nm–2 µm) (Figure 2) [85,86]. The reason why EVs can communicate with each other is they carry many useful active substances, including lipids, protein [87], and nucleic acid substances such as DNA, mRNA, miRNA, or long-chain non-coding RNA [88,89], which can effectively reflect the state of donor cells [90,91]. This biological information is carried by EVs and participates in the transmission of information between cells. They carry out a series of intercellular substance uptake, transport, and transfer processes and then enter into various physiological processes between cells in the body [92], for example, the exchange of nucleic acids and proteins between cells, the immunomodulatory processes in the body, and the induction of angiogenesis [93,94,95]. As nanocarriers, EVs are endowed with cell-based biological structures and functions during the process of drug delivery, giving them inherent intercellular communication capabilities, enhanced cell-to-cell communication, and greater chemical stability. Moreover, EVs can achieve similar effects to synthetic nanocarriers like liposomes. Currently, widely studied drug carriers such as liposomes or polymer-based carriers are prone to phagocytosis by hepatic and splenic macrophages during in vivo circulation. Additionally, these carriers suffer from drawbacks such as short circulation time, poor stability, and low targeting specificity in the bloodstream [96]. The discovery of EVs ingeniously addresses these inevitable shortcomings of nanocarriers. Compared to synthetic nanocarriers, EVs can avoid being engulfed or degraded in circulation [97]. Being naturally secreted substances, they are able to overcome natural barriers such as the blood–brain barrier and are inherently stable when circulating in the recipient [98,99,100], and their immunogenicity is also lower than other traditional carriers [101]. Stem-cell-derived EVs exhibit inflammatory tropism, enabling them to exert therapeutic effects. In addition, EVs can achieve more precise drug targeting and delivery due to their complex and unique membrane structure [102]. Compared to synthetic nanocarriers, they can accommodate a greater variety of non-biological biomimetic materials, providing them with greater medical value. Therefore, as drug delivery vehicles, EVs possess natural advantages that make them safer, more stable, and more precise than other synthetic nanocarriers [103].

### 4.2. Isolation of EVs

Although many studies have proved the feasibility and safety of EVs as drug delivery carriers, the standards and requirements for the clinical application of them are extremely high. A series of preparations for EV production should be carried out, including increasing production, isolation and purification, and so on, before being used as a drug carrier in clinical trials [104]. Differential ultracentrifugation is one of the most frequently used methods for the isolation of EVs. After granulation via differential ultracentrifugation using high gravity, further purification was carried out using a sucrose gradient [105]. But this kind of method is not applicable in clinical settings because it requires a high time cost, and the operation may lead to rupture [106]. In addition, there are two types of separation methods. The first approach is immunoaffinity isolation based on the selective capture of EVs expressing specific surface biomarkers, but it is currently unclear whether specific subtypes are more or less feasible for drug delivery purposes. Another separation method underlies the size of the EVs themselves [107]. The corresponding limitation is that binding non-specific EVs proteins to cell membranes can lead to low recovery rates. However, the defects of these two methods also make them limited in clinical practice. A new method of using size exclusion chromatography involves separating EVs from plasma, allowing for a higher standard of isolation and purification that can be used in clinical applications for the purpose of drug delivery [108]. Although there are many separation techniques, the heterogeneity in size and composition creates an overlap in the size and morphology of different subtypes of EVs, which leads to the inability to completely purify and separate a population [85,109]. Therefore, there is currently a lack of standardized protocols for achieving the accurate isolation and identification of EVs in clinical applications. In order to discuss the difficulties in extraction and low production caused by heterogeneity, MNVs artificially mimicking EVs could be used to replace them for disease treatment [110]; they can be formed in any cell, have highly similar characteristics to EVs, and have the same function of transmitting information. Moreover, they can be produced in high quantities. For example, MNVs can be produced 100 times or more than exosomes [111,112]. Thus, nanocarriers inspired by EVs for targeted drug delivery can be divided into (1) isolated and purified EVs naturally occurring in human body and (2) simulators of EVs, namely MNVs [113]. The advantages and disadvantages of these two types of EVs as nanocarriers are summarized in Table 1.

### 4.3. EV Modification

Modification can endow EVs with additional characteristics such as the ability to escape from immune detection and effective accumulation at the target site. It is a key step in establishing a nanocarrier targeted drug delivery system that increases cycle time and enhances intracellular uptake [114]. EVs secreted by stem cells can be used in drug delivery systems to transform membrane carriers suitable for drug delivery through modifying the cell molecules contained in them, modifying the membrane, or fusing the outer vesicle membrane with other cell membranes [115].

The content of transformation involves the delivery of drugs to the target organ. It can be divided into endogenous and exogenous loading according to the source of the delivered substances [116,117]. The first way is endogenous loading, which involves integrating the therapeutic contents into the vesicles before the formation of the outer vesicles and directly generating the processed drug-loaded vesicles before the whole organism develops. In other words, the drugs are loaded into the donor cell first. When the drugs are sorted into the EVs and released from the donor cell, the drug-loaded vesicles are obtained through separation and purification. The second technique, exogenous loading, is to load the therapeutic molecules that need to be delivered into the EVs that have been separated and purified by suitable means (Figure 3A) [118]. The surface modification of Evs is the most commonly used method. This method can effectively reduce drug toxicity and make the biological function of nanovesicles more stable, such as polyethylene glycol modification, overexpression of CD47, peptide modification, etc. (Figure 3B) [119,120,121,122]. In addition, it can also be fused with other kinds of cell membranes such as platelets. This naturally occurring membrane fusion can maintain the integrity of the phospholipid bilayer while they are fusing, so as to achieve the purpose of biological camouflage and avoid immune clearance (Figure 3C) [123].

### 4.4. Natural EVs

#### 4.4.1. Exosomes

Exosomes are the smallest vesicles of relatively uniform size at the nanoscale, ranging in size from 40 to 150 nm [124,125]. Their formation originates from the budding of multivesicular endosomes to form early endosomes, followed by late endosomes and the subsequent formation of intracavitary vesicles to form multivesicular bodies that fuse with the plasma membrane to form and release products. This obtained product is the exosome [82,126]. It plays an important role in material transportation and signal transmission in the body. The membranes of exosomes are phospholipid bilayers which have stable characteristics. This kind of membrane structure can effectively participate in the above-mentioned endocytosis process [127]. It can also participate in the material exchange between cells at the later stage that plays a key role in intercellular communication [128]. The surface of an exosome is inlaid with a variety of protein substances, such as integrin protein, quadruple transmembrane protein, polysaccharide protein, and other substances [129]. Like all released vesicles, exosomes carry a variety of substances such as RNA, proteins, and lipids inside. These bioactive substances are transferred to recipient cells by exosomes during the processes of cell communication and intercellular conduction, so that some genetic pathological expressions can be modified, and even gene editing can be carried out. Exosomes are widely found in various body fluids, such as saliva, urine, blood, and so on. Not only that, they can be produced in a variety of cells such as immune cells, epithelial cells, and stem cells. Among these, the number of exosomes secreted by mesenchymal stem cells is the highest [124,130,131]. Thanks to their structure and microenvironmental distribution, exosomes have shown powerful functions in regulating gene transcription and translation, cell proliferation and apoptosis, angiogenesis, immune response, and viral immunity.

The exosomes secreted by stem cells have various protective effects on the body, especially the exosomes derived from BMSCs. They are the main force of cells in secretory membrane vesicles, including exosomes [132,133,134]. When BMSCs are used as cells to treat diseases, they cannot reach the best expected value in clinical trials. Under these circumstances, the exosomes secreted by them have become the best alternative to replace BMSCs in the treatment of diseases or the targeted delivery of drugs [135]. As a paracrine substance, exosomes have properties similar to those of BMSCs, but also have advantages that BMSCs do not have. As one type of EVs, exosomes can also play a good role in treating various pathologies in clinical research [136,137]. Exosomes derived from various types of stem cells have demonstrated significant therapeutic potential, such as in anti-tumor therapy. Relevant studies have shown that when paclitaxel is transfected into EVs secreted by embryonic stem cells (ESCs), it substantially enhances the targeting effect and efficacy of paclitaxel in glioblastoma [138]. In addition, exosomes are one of the features and tools for regulating them in terms of cardiac function, such as treating heart infarction and myocardial injury [139,140].

The reason why exosomes have aroused research enthusiasm in terms of drug delivery is that they have more potential and advantages to be drug carriers than traditional nanoparticles or stem cells. Exosomes are not only similar to the structure and function of cells but also can achieve a similar effect to traditional nanoparticles in drug delivery [141,142]. As nanoscale vesicles, exosomes are similar to nanoparticles that are capable of loading and transporting large amounts of active contents to recipient cells for targeted transport and sustained release. And on this basis, exosomes can also be made into specialized carriers, which complements their innate targeting and messaging capabilities [143]. Exosomes have become a safer and more natural tool for clinical use than nanoparticles and stem cells [144]. Using exosomes as nanocarriers for drug delivery can give greater play to the targeting of exosomes. Compared with traditional nanocarriers, the bioavailability of exosomes will be higher, and the toxicity and side effects will be greatly reduced [145]. But it must be mentioned that the extraction and purification of exosomes in large quantities remains a difficult challenge even with such broad therapeutic advantages [146]. This is a barrier to the clinical conversion of exosome therapy. The separation, purification, and characterization of specific exosomes require a lot of complicated processes, and the efficiency needs to be improved [147]. The heterogeneity of exosomes makes this problem more complex and difficult, which requires a lot of research and innovation to break through this obstacle [148,149]. The research on exosomes has been more extensive, but the development of the other two types of EVs seems to be relatively stagnant. However, it is noteworthy that microvesicles and apoptotic bodies also have great potential in drug delivery and disease treatment.

#### 4.4.2. Microvesicles

Ectosomes and shedding vesicles are terms that may also refer to microvesicles, which contain growth factors and genetic material that help send signals to target cells [10]. Although the occurrence of microvesicles and exosomes equally involve membrane transport, the biogenic process of the former is different from the complex production pathway of the latter. Their unique feature lies in its direct discharge from the plasma membrane and their formation from the germination and exfoliation of the plasma membrane [150]. It can be inferred from the different processes that microvesicles are more likely to have a plasma membrane highly similar to their donor cells and show the same selectivity and targeting as their donor cells. This replication may be a special feature that distinguishes them from exosomes in the process of targeting delivery [151]. And the volume of microvesicles is larger than that of exosomes, ranging from 100–1000 nm. And the inclusions of both may serve as the key to distinguish them, e.g., the proteins of the exosomes are specific, and they also have an asymmetric phosphatidylserine [152]. Microvesicles can collect specific proteins, such as transmembrane proteins which are secreted by a variety of cells including stem cells, and are found in various body fluids in the body because of their continuous divergence. They are also an important tool for intercellular communication in the body due to their multiple cell sources and divergent characteristics.

It has been observed that released microvesicles have the characteristics of acting on target cells with specific receptor ligands rather than randomly binding with arbitrary cells [153]. Not only that, as secreted membrane vesicles, they carry genetic material such as mRNA and microRNA, which can be transferred into the target cells and have an impact on the characteristics of the target cells. For example, microvesicles can be used as carriers to transport microRNA to the recipient cells to reduce ischemic myocardial injury [154]. It can be understood that microvesicles not only have the function of information exchange between each other and specific binding to the target cell, but also can transport the gene product carried by the maternal cell to the recipient cell. It can be used as a medical diagnostic tool by means of editing receptor cell genes through this feature and specifically identifying cell sources when microvesicles participate in the human pathological process [155]. In disease therapy, stem-cell-derived microvesicles have properties similar to stem cells, but they can avoid the safety and technical issues arising in stem cell therapy at the same time. It can be said that microvesicles can effectively replace stem cell products in medical treatment as a natural paracrine substance of stem cells, and can effectively avoid various side effects or adverse reactions [156]. In addition, stem cells can also act as “mediators” between cells and balance the microenvironment through the intercellular communication conduction in the microenvironment, so as to reduce the damage caused by disease to the human body [157,158]. For example, in the context of cardiovascular disease treatment, microvesicles derived from iPSCs can demonstrate protective properties through signal transduction, thereby reducing inflammatory responses [159]. Apart from this, microvesicles exhibit potential involvement in the treatment of tissue damage, cardiovascular and kidney diseases, and tumors [160].

According to reports, microvesicles can directly stimulate receptor cells to complete signaling and can also serve as particle carriers while delivering information to the target [161]. The rich transmembrane proteins on the surface of the vesicles, as well as the active substances inside the vesicles, ensure efficient communication between cells and the capacity to load a large number of carriers. It lays a strong foundation for the new research field of microvesicles as a drug delivery system [162]. And they have inherent stability and immune tolerance, as the membrane vesicles produced by the budding of human cell membranes, when compared to traditional drug-loading systems. However, synthetic traditional nanocarriers will face difficult problems such as low biocompatibility and insufficient stability in the environment [163]. In addition, under the limitation of cell therapy, microvesicles produced by cells can be used as biomimetic nanocarriers to transport cargo to target cells, thus achieving certain therapeutic effects and goals. It can be seen from this comparison that they are ideal natural drug delivery carriers [164].

#### 4.4.3. Apoptotic Bodies

Apoptotic bodies are the largest class of vesicles, which are about 500 nm–2 µm in diameter. The formation of them is one of the typical features of apoptosis, which is induced through the process of cellular transcription. Apoptosis is a very complex and active process [165,166]. The initial event during cellular apoptosis involves the condensation of nuclear chromatin. Subsequently, fission occurs, followed by the emergence of micronuclei and membrane vesicles. Apoptotic bodies are the division of the nuclear contents into different membrane-coated vesicles [167]. Subsequently, they are engulfed by various types of cells, including epithelial cells, macrophages, and others, and are ultimately ingested and degraded in the lysosome [168]. Apoptotic bodies appear upon cell lysis into debris, which usually occurs in the final phase of cell-programmed apoptosis; thus, the formation of apoptotic bodies can also be regarded as the final marker of apoptosis [169]. Apoptotic bodies contain a variety of intact or incomplete and already-apoptotic intracellular residual components due to their formation pathway, including part of the cytoplasm, microRNAs, mRNA, DNA fragments, proteins, etc. This distribution is random and uneven [170].

Nowadays, the role of intercellular communication in EVs is widely studied. It shows the potential of apoptotic bodies with greater contents and that the possibilities of richer morphological changes in the outer vesicle group on human cells may be unlimited [171]. It has been found that intracellular molecules can be transferred to recipient cells by apoptotic body packaging, thereby mediating communication expression within the receptor cells. For example, after the infusion of exogenous apoptotic bodies into BMSCs with impaired autogenetic and bone\adipogenic differentiation due to insufficient apoptosis, the Wnt\β-catenin pathway is activated through RNF146 and miR-328-3p in apoptotic cells, thus maintaining damaged MSC characteristics [172]. This suggests that apoptotic bodies can exist as carrier vesicles for the material. Through using this property, we might like to speculate that apoptotic bodies have transport communication functions similar to exosomes and microvesicles; perhaps they could be a new tool for nanoscale drug loading [173,174]. However, the current stage of apoptotic bodies has not been deeply studied, the mechanism of action is not clear, and more far-reaching research and exploration are needed [175].

### 4.5. Simulators of EVs: Mimetic Nanovesicles (MNVs)

Research usually uses EVs naturally secreted by cells in the body as a means of providing drug therapy. However, there are limitations in treatment due to issues such as low production and purification differences. There is an urgent need for an expanded and modified loading method based on this in an effort to overcome this challenge and expand the scope and depth of treatment. Therefore, improved MNVs with the same unique characteristics have been derived [176]. There is an EV that is not involved in physiological mechanisms in the body and is artificially produced in vitro [110]. It is a type of nanovesicle that can be designed using any cell type or produced through fusion with prerequisite materials such as liposomes [177]. The production method is to use nanosized filters with reduced pore size to continuously compress and decompose cells, and separate them from the interface between the 10–50% iodixanol layers through a two-step density gradient ultracentrifugation [178]. Drugs can be loaded during the extrusion process, which means eliminating additional loading steps and procedures. It makes the loading method simpler [179]. It has also been used as a replacement for nanocarriers in treatment because it mimics the biological characteristics of natural EVs found in the human body [180]. Additionally, it can even surpass EVs due to its artificial design, which allows it to load goods and target specific cells or tissues to improve uptake. More importantly, it has a simpler separation and drug loading procedure compared to EVs, and the production can be 100 times higher [181]. The prostate cancer chemotherapy drug docetaxel is loaded into MNVs prepared via the extrusion of iPSCs-MSCs for subcutaneous and bone metastatic prostate cancer mouse models. Through observing the growth status of tumors, it was found that the drug treatment effect was significantly enhanced, and the treatment of metastatic prostate cancer was effectively improved [180]. From this, it can be seen that MNVs can deliver targeted tumor drugs efficiently. They are a promising targeted drug delivery carrier that can improve the treatment efficiency of diseases.

However, corresponding challenges also arise. Firstly, the extrusion production method may cause membrane deformation. Secondly, the process of extracting components and fusing with liposomes during production may be more difficult and complex than theoretically, which would result in lower overall satisfaction [178]. Finally, as a result of its origin in cells, the issue of vesicle heterogeneity cannot be completely overcome and avoided. Therefore, it is particularly crucial to improve the separation and purification technology of vesicles in future development. And it is also necessary to conduct in-depth research on the preparation and operation mechanism of MNVs to create a broader new platform for targeted drug delivery.

## 5. Application of Stem Cell EVs in Disease Treatment

### 5.1. Cardiovascular Diseases

Cardiovascular disease is a category of diseases that seriously affect human life and health, such as hypertension, coronary heart disease, mental infarction, etc. At present, the treatment of cardiovascular disease is mainly precision treatment. EVs as natural nano-carriers could allow for the targeted delivery of therapeutic agents to recipient cells, so as to achieve accurate drug administration. Zhao et al. discovered that EVs derived from MSCs can alleviate myocardial ischemia/reperfusion (I/R) injury in mice through modulating the polarization of M1 macrophages towards M2 macrophages. This crucial macrophage polarization is achieved through the involvement of miR-182, an important candidate mediator [182]. Other researchers have found that intracoronary injection of EVs derived from cardiosphere-derived cells (CDCs) carrying microRNA is safe and can improve cardiac function in a pig model of dilated cardiomyopathy (DCM) [183]. Additionally, in a mouse model of dilated cardiomyopathy, intravenous injection of exosomes derived from MSCs significantly reduces the number of pro-inflammatory macrophages in the blood and heart. This effect is attributed to the ability of exosomes to regulate macrophages and alleviate inflammation in the cardiac microenvironment [184]. Sun et al. demonstrated the role of EVs derived from MSCs overexpressing HIF-1α in promoting neovascularization and inhibiting fibrosis to maintain cardiac function in a rat model of myocardial infarction (MI) [185]. In addition, other studies have revealed that exosomes derived from mouse ESCs can enhance neovascularization, promote the survival of cardiomyocytes, reduce fibrosis after infarction, and restore cardiac function in a mouse model of acute myocardial infarction. This study indicates that this ability is specifically associated with miR-294 [186].

### 5.2. Neurodegenerative Diseases

Neurodegenerative diseases are chronic, insidious, and progressive disorders that occur in the central nervous system (CNS). They are characterized by the loss of neuronal structure and function, leading to impaired communication between neurons [187]. The complexity of the disease and the lack of comprehensive understanding of its specific pathological mechanisms have resulted in significant shortcomings in current research. Currently, numerous studies have demonstrated the clear therapeutic effects of stem-cell-derived EVs in promoting neurovascular regeneration, modulating inflammatory environments, and more [188]. Injecting human neural stem cell (hNSC)-derived EVs intravenously into early- or late-stage AD mouse models, experimental results demonstrate the neuroprotective effects of stem-cell-derived EVs in repairing both behavioral and molecular AD neuro-pathologies [189]. Ding et al. injected exosomes derived from human umbilical cord MSCs (hucMSCs) into the bodies of AD model mice and observed the restoration of cognitive impairment in these mice. Furthermore, it was discovered that the exosomes could regulate the activation of microglial cells in the mouse brain, thereby alleviating neuroinflammation [190]. Parkinson’s disease (PD) is a neurodegenerative disease with a high incidence rate, second only to AD [191]. The more reasonable way to treat Parkinson’s disease is to use catalase. However, catalase cannot pass through the brain barrier, so it cannot directly reach the target neuron area to be treated [192]. Using the characteristics of exocrine bodies that can penetrate the blood–brain barrier, catalase was loaded into exosomes and tested in a Parkinson’s mouse model. The results showed that the preparation could penetrate the brain barrier and reach the corresponding lesion area, thus effectively alleviating the disease [193]. Xin et al. discovered that exosomes derived from MSCs can promote neurovascular remodeling and functional recovery in a rat model of stroke [194].

### 5.3. Liver Diseases

Liver disease is the main global health problem leading to death and disability, with approximately 2 million deaths worldwide each year [195]. Yan et al. confirmed that extracellular vesicles derived from hucMSC can alleviate oxidative stress and cell apoptosis through delivering derived glutathione peroxidase 1 (GPX1) to detoxify CCl_4_ and H_2_O_2_, thereby promoting the recovery of hepatic oxidative injury [196]. MiR-122 target genes are involved in the proliferation of hepatic stellate cells (HSCs) and the maturation of collagen. The modification of MiR-122 improves the therapeutic effect of adipose tissue-derived MSCs (AMSCs) on CCl_4_-induced liver fibrosis through inhibiting HSC activation and reducing collagen deposition. It is a promising strategy for the treatment of liver fibrosis [197].

### 5.4. Lung Diseases

Lung disease is a prevalent global health concern and a leading cause of illness and mortality worldwide, including chronic obstructive pulmonary disease, influenza and other respiratory infections, acute respiratory distress syndrome, asthma, and pneumonia–cancer [198]. Research has revealed that the antiviral activity of MSC-EVs is attributed to the transfer of RNA from EVs to epithelial cells. In a pig model of influenza virus infection, intratracheal injection of MSC-EVs resulted in a significant reduction in the release of the virus in nasal swabs taken from infected pigs at 12 h post-infection. The replication of the influenza virus was markedly inhibited, and the expression of virus-related pro-inflammatory cytokines was downregulated. Histopathological findings demonstrated that MSC-EVs reduced lung damage caused by the influenza virus in pigs [199]. Mansouri et al. investigated the therapeutic effects of MSC-EVs in a bleomycin-induced pulmonary fibrosis model. The study found that MSC-EVs were able to prevent and reverse the core features of bleomycin-induced pulmonary fibrosis [200]. Furthermore, MSC-EVs have been found to improve disease-associated mitochondrial dysfunction in both a hypoxia-induced mouse model and a semaxanib/hypoxia rat model of pulmonary arterial hypertension (PAH). These findings highlight the therapeutic potential of EVs in the treatment of PAH [201]. Since the end of 2019, Coronavirus disease-19 (COVID-19) has become a new global threat that can lead to compromised immune systems, exacerbate the production of inflammatory cytokines, and cause coagulation disorders. According to reports, miRNA carried by MSC-derived EVs (microvesicles and exosomes) can alleviate inflammatory factors, prevent tissue damage, and counteract coagulation disturbances. Therefore, MSC-EVs-miRNA has become a potential multi-target treatment approach for COVID-19 [202].

### 5.5. Kidney Diseases

The number of acute kidney injuries (AKIs) and chronic kidney diseases (CKDs) is soaring around the world, and treatment for the disease is only slowing down [203]. MSCs can repair the kidney because they can increase the proliferation of renal tubular cells and effectively inhibit apoptosis. The EVs secreted by it have now replaced cell therapy, reduced the inflammatory reaction in different acute or chronic kidney injury models, and promoted tissue repair and regeneration, playing a protective role. In a mouse model of acute renal injury induced by glycerol, the EVs secreted by bone marrow mesenchymal stem cells promote the proliferation of renal tubular cells and repair the damaged renal tubular cells. Through carrying specific mRNA, the damaged renal tubular cells are stimulated to proliferate, and the damaged cells re-enter the cell cycle [204], effectively improving renal function. In a chronic cyclosporine (CsA)-induced nephrotoxicity invasive mouse model, BMSCs, their EVs, and EV-depleted conditioned medium have shown renal protective effects, improving the prognosis of kidney disease [205].

As described above, extensive studies have demonstrated the potential clinical applications of EVs (Table 2). However, these approaches are still in the early stages of research and clinical application, and there are still several challenges to be considered when transitioning from the laboratory to clinical practice. Standardization and improvement of the isolation and storage of EVs, as well as the development of quantitative, stable, and reproducible assays to assess their therapeutic potential, are essential to facilitate the translation of EVs from the laboratory to clinical applications.

## 6. Conclusions and Future Prospectives

Stem cells show their strong therapeutic potential for clinical treatment. Even though stem cells are not a class of clinical drugs yet, stem-cell-based therapies bring new prospects in the field of medicine [206]. Stem cell therapy has been extensively studied in a variety of diseases and trauma studies, and most of them have yielded encouraging results. However, the ability of stem cells is limited in terms of migration and homing, creating obvious barriers to clinical translation. Biomimetic nanotechnology has attracted attention in recent years, which becomes a promising therapeutic strategy [207]. Stem cells and secreted EVs have come into focus and become ideal natural materials. The membranes of stem cells control the flow of substances coming in and out, and they are also responsible for the exchange of information between cells. Stem cell membrane-coated nanomaterials can replicate the natural characteristics of cells onto nanomaterials, displaying advantages such as high targeting and reducing side effects and toxicity [208]. However, their safety and effectiveness still need to be proven through a large amount of clinical practice information, and the current research is still at a relatively superficial stage. Their stability and production complexity are challenges that need to be confirmed through more in-depth research [209]. EVs secreted by stem cells act as carriers of intercellular biological information, and different vesicle types make a great contribution to cell communication [210,211]. As a secretome of stem cells, EVs can be used for the diagnosis and treatment of diseases with their unique structure and rich contents. And they have caused great interest among researchers and developed greatly in recent years. Numerous research findings have indicated that EVs can serve as potential targeted therapy in disease treatment, and this new approach has been scientifically accepted. Therefore, the secretomes of these stem cells, EVs, have become a research hotspot and a choice for disease treatment as a novel therapeutic tool. In future research, these secretomes may serve as a breakthrough in discovering new disease targets. At the same time, they have also made a great contribution to the nano-loaded drug. EVs are natural nanomaterials, and their unique physicochemical properties allow them to serve as unique drug carriers [212]. EVs possess cellular tropism that traditional drug carriers lack, which not only enhances drug targeting but also improves efficiency and safety [213]. Its low immunogenicity and other characteristics can effectively avoid the disadvantages of traditional drug loading [98,214]. Loading therapeutic drugs into biologically inspired vesicles to deliver drugs to form a targeted drug delivery system is expected to replace stem cell therapy as a novel treatment method, and it is also a promising approach for cell-free therapy. Thus, Evs are natural tools for communication and drug loading, and exhibit the most potential and promise in drug-carrying technology. However, the purification process of outer vesicles and mass production technology are the obstacles that need to be improved and solved at the present stage. If these issues can be addressed in clinical development and the drug delivery mode of nanovesicles with high safety and high yield can be explored, there will be broad application prospects in drug delivery systems and the clinical treatment of diseases. The EVs of stem cells should be further studied to lay a solid foundation for medical research. It is certain that the use of this new targeted drug carrier will have a significant positive impact on human health, becoming a promising means of disease control and treatment.

## Figures and Tables

**Figure 1 pharmaceutics-15-02011-f001:**
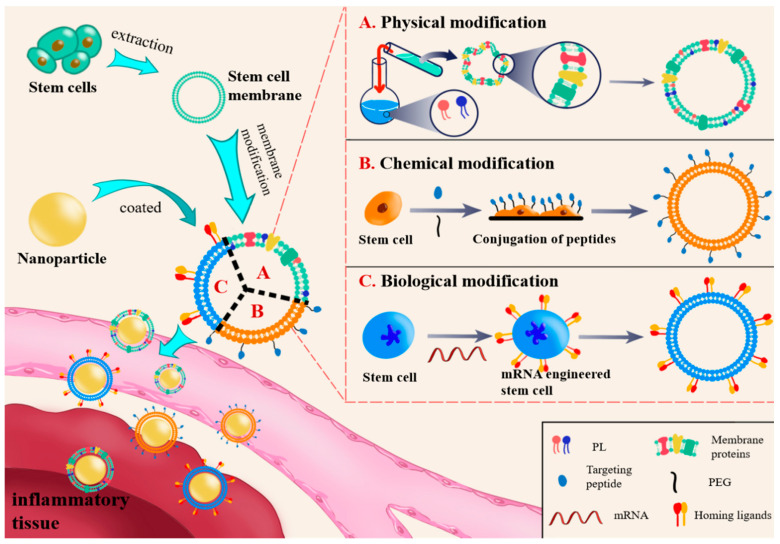
The cell membrane extracted from stem cells is modified and loaded with nanoparticles for transmission to the inflammatory site. There are three methods for cell membrane modification: (**A**) physical modification—the phospholipid materials are anchored onto the membrane; (**B**) chemical modification—coupling targeted peptides onto stem cell membranes; (**C**) biological modification—mRNA engineered stem cells expressing a combination of homing ligands on the cell membrane surface.

**Figure 2 pharmaceutics-15-02011-f002:**
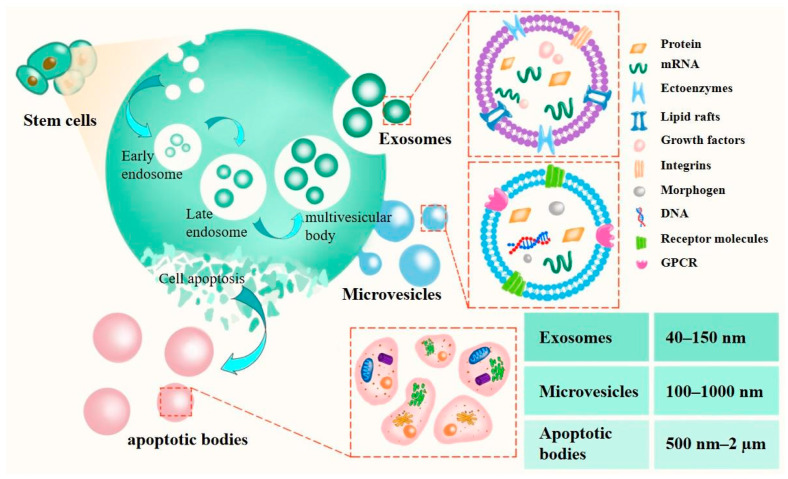
The secretion process of three types of EVs. The formation of exosomes originates from the formation of early endosomes through the germination of multi-vesicular endosomes, followed by late endosomes; then, it forms multivesicular bodies which are fused with the plasma membrane and released outside the cell. Microvesicles are direct discharge from the plasma membrane, and they are from the germination and exfoliation of the plasma membrane. Apoptotic bodies are produced when cells enter an apoptotic state.

**Figure 3 pharmaceutics-15-02011-f003:**
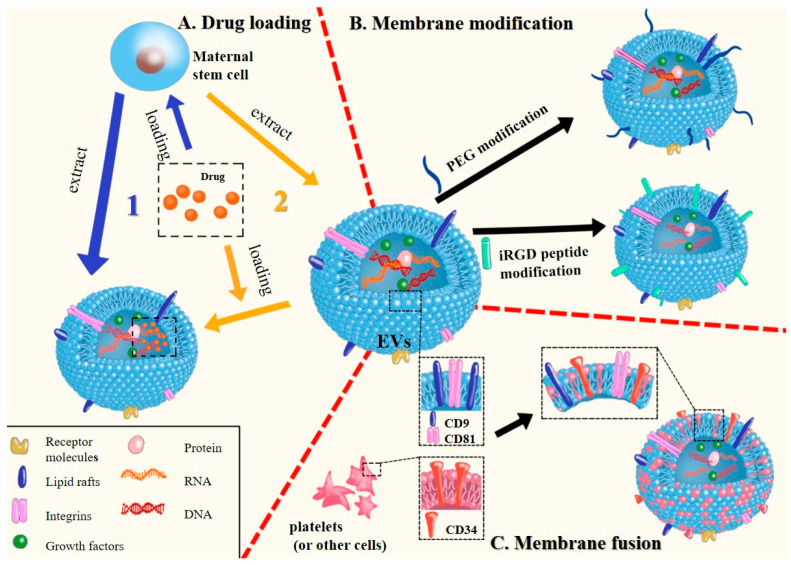
Modification of EVs to establish a nano-drug delivery system: EVs collected from stem cells can be utilized in drug delivery systems to create membrane carriers suitable for drug delivery through the following methods. (**A**) Drug loading methods: 1. Loading drugs into maternal stem cells and extracting EVs containing drugs. 2. Directly incubating extracted and separated EVs with drugs, allowing the drugs to diffuse into the EVs. (**B**) Membrane modification: Modifying the membrane surface with PEG or targeted peptides. (**C**) Membrane fusion: This is a naturally occurring process where EV membranes can fuse with other types of cell membranes, such as platelets. The fusion of these membranes can occur without leakage or loss of lipid bilayer integrity.

**Table 1 pharmaceutics-15-02011-t001:** Advantages and disadvantages of stem-cell-derived EVs and simulators of EVs (MNVs).

	Advantages	Disadvantages
Stem cells-derived EVs	Wide range of sources	Lack of clinical translation standards
	Intercellular communication abilities	Challenges in separation and purification
	Avoidance of phagocytosis or degradation in circulation	Low yield, hindering large-scale production
	Low immunogenicity	Optimal storage protocols and quality control require further investigation
	No cellular toxicity	Lack of understanding of the underlying mechanisms
	Ability to overcome natural barriers	
	High chemical stability	
	Inflammatory tropism and targeting specificity	
	Good biocompatibility	
simulators of Evs (MNVs)	High yield	Inability to completely avoid heterogeneity
	Simple preparation process	Limited achievement of efficient cargo loading
	Simple drug-loading method	Comprehensive research on safety, systemic immune response, and efficacy is still lacking

**Table 2 pharmaceutics-15-02011-t002:** Application of stem cell-EVs in disease treatment.

Disease	Cell Source	Human\Animal Model	Cargo/Drug	Reference
myocardial I/R injury	MSC-EVs	mice	miR-182	[182]
DCM	CDC-EVs	pig	microRNA	[183]
MI	MSC-EVs	rat	HIF-1α	[185]
MI	ESC-exosomes	mice	miR-294	[186]
AD	hNSC-EVs	mice	miRNA	[189]
PD	exosomes	mice	catalase	[193]
Stroke	MSC-exosomes	rat	miRNA	[194]
Hepatic Oxidant Injury	hucMSC-exosomes	mice	GPX1	[196]
Liver fibrosis	AMSC-exosomes	mice	miR-122	[197]
Acute lung injury	MSC-EVs	pig	mRNA	[199]
Pulmonary fibrosis	MSC-exosomes	mice	noncoding RNA	[200]
COVID-19	MSC-exosomes and microcesicles	patients	miRNA	[202]
AKI	MSC-microvesicles	mice	mRNA	[204]
Chronic CsA nephrotoxicity	MSC-EVs	mice	CD44	[205]

I/R = ischemia/reperfusion; DCM = dilated cardiomyopathy; MI = myocardial infarction; AD = Alzheimer’s disease; PD = Parkinson’s disease; COVID-19 = Coronavirus disease-19; AKI = acute kidney injury; CsA = chronic cyclosporine; EVs = extracellular vesicles; MSC = mesenchymal stem cell; CDC = cardiosphere-derived cell; ESC = embryonic stem cell; hNSC = human neural stem cell; hucMSC = human umbilical cord MSC; AMSC = adipose tissue-derived MSCs.

## Data Availability

Not applicable.

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
