# Peer review of "Bio-Inspired Nanocarriers Derived from Stem Cells and Their Extracellular Vesicles for Targeted Drug Delivery"

_pharmaceutics, 2023, doi:10.3390/pharmaceutics15072011_

Round 1
Reviewer 1 Report
In general, this review article provides a clear and comprehensive description of the current knowledge on targeted drug delivery using stem cells and their extracellular vesicles as biomimetic nanocarriers. The manuscript represents an important update on the topics of extracellular vesicle and drug delivery, and is likely to generate broad interest among researchers working in the field. This manuscript is well-constructed, the literature is carefully described, and the various issues are presented in a well-organized manner. Couple of minor issues need to be addressed to bring this manuscript to a publishable level:
1. To gain better understanding on the potential applications of extracellular vesicles, please provide a specific discussion on their usage as delivery vehicles for treating different diseases.
2. Further discussion is needed regarding the specific biological modifications on the membrane surface.
3. The discussion on the benefits of stem cell-based nanocarriers need to be further clarified. For example, it would be helpful to explain why stem cell membranes tend to accumulate in inflamed tissues and exhibit good immune escape ability.
4. When discussing the benefits of stem cells and their extracellular vesicles, it would be beneficial to compare them with other nanomaterials.
5. In Figure 3, the mentioned surface markers (e.g., CD9, CD81, and CD34) are transmembrane proteins with shorter or longer cytosolic domains. The current illustration gives the impression that they are anchored on the membrane surface and penetrate only one lipid layer. Please revise the illustration accordingly.
6. It would be helpful to include a list of abbreviations.
English is good.
Author Response
Dear Ms. Clarissa Chen
Thank you for your email dated 4 July 2023.
We have taken into consideration the constructive comments and suggestions of the reviewers for our manuscript, and have revised our manuscript accordingly. The corrections are highlighted in red in the revised manuscript.
Please find attached our revised manuscript, as well as our point-by-point response to the reviewers.
Again, thank you for taking the time to review this manuscript. We trust the present manuscript meets with the standards of Pharmaceutics.
Looking forward to hearing from you soon.
Yours sincerely, Zhi-xiang Yuan
We would like to express our sincere thanks to the editor and reviewers for the constructive and positive comments.
Response to the reviewers:
Reviewer 1
In general, this review article provides a clear and comprehensive description of the current knowledge on targeted drug delivery using stem cells and their extracellular vesicles as biomimetic nanocarriers. The manuscript represents an important update on the topics of extracellular vesicle and drug delivery, and is likely to generate broad interest among researchers working in the field. This manuscript is well-constructed, the literature is carefully described, and the various issues are presented in a well-organized manner. Couple of minor issues need to be addressed to bring this manuscript to a publishable level:
1.To gain better understanding on the potential applications of extracellular vesicles, please provide a specific discussion on their usage as delivery vehicles for treating different diseases.
A: Thanks a lot for reviewers’ suggestion. We have added the section 5. Application of stem cell-EVs in disease treatment:“5.1 Cardiovascular diseases
Cardiovascular disease is a category of diseases that seriously affect human life and health, such as hypertension, coronary heart disease, mental infarction, etc. At present, the treatment of cardiovascular disease is mainly precision treatment. EVs as natural nano-carriers, could allow for the targeted delivery of therapeutic agents to the recipient cells, so as to achieve accurate drug administration. Zhao et al. discovered that EVs derived from MSCs can alleviate myocardial ischemia/reperfusion (I/R) injury in mice by modulating the polarization of M1 macrophages towards M2 macrophages. This crucial macrophage polarization is achieved through the involvement of miR-182, an important candidate mediator[182]. Other researchers have found that intracoronary injection of EVs derived from cardiosphere-derived cells (CDCs) carrying microRNA is safe and can improve cardiac function in a pig model of dilated cardiomyopathy (DCM)[183]. Additionally, in a mouse model of dilated cardiomyopathy, intravenous injection of exosomes derived from MSCs significantly reduces the number of pro-inflammatory macrophages in the blood and heart. This effect is attributed to the ability of exosomes to regulate macrophages and alleviate inflammation in the cardiac microenvironment[184]. Sun et al. demonstrated the role of EVs derived from MSCs overexpressing HIF-1α in promoting neovascularization and inhibiting fibrosis to maintain cardiac function in a rat model of myocardial infarction (MI)[185]. In addition, other studies have revealed that exosomes derived from mouse ESCs can enhance the survival of neovascularization and cardiomyocytes, reduce fibrosis after infarction, and restore cardiac function in a mouse model of acute myocardial infarction. The research indicates that this ability is associated with specific miR-294[186].” ... “As described above, extensive studies have demonstrated the potential clinical applications of EVs (Table 2). However, these approaches are still in the early stages of research and clinical application and there are still several challenges to be considered when transitioning from the laboratory to clinical practice. Standardization and improvement of the isolation and storage of EVs, as well as the development of quantitative, stable, and reproducible assays to assess their therapeutic potential, are essential to facilitate the translation of EVs from the laboratory to clinical applications.”
- Further discussion is needed regarding the specific biological modifications on the membrane surface.
A: As suggested by reviewers, We have added discussion on specific biological modifications on the membrane surface:“By modifying stem cell membranes, it is possible to create membrane-coated nanoparticles with specialized functionalities that go beyond what the cell membrane alone can provide[46]. The modified membrane allows the encapsulated nanoparticles to possess the complex and distinctive surface physicochemical properties of stem cells, thereby extending their blood circulation time, enhancing active targeting, and improving cellular internalization. This nanocarrier system can address challenges associated with biocompatibility, immune responses, and off-target effects caused by nanoparticles[47].”
- The discussion on the benefits of stem cell-based nanocarriers need to be further clarified. For example, it would be helpful to explain why stem cell membranes tend to accumulate in inflamed tissues and exhibit good immune escape ability.
A:According to reviewers’ comment, we have included the description of the benefits of stem cell-based nanocarriers: “Stem cells exhibit inflammation-driven tumor tropism mediated by adhesive ligands such as Sialyl Lewis X (SLeX) and P-selectin glycoprotein ligand-1 (PSGL-1). The presence of surface antigens and innate targeting ligands greatly favors their use as tumor-targeted carriers. Currently, stem cells have been employed for the delivery of protein and peptide-based anticancer drugs, such as interferons and interleukins. Stem cell nanocarriers are highly favored for their safety profile in terms of non-oncogenicity[36]. In recent years, stem cell membrane has been used as a natural coating material taking advantages of core nanoparticles and source cells, in order to achieve highly efficient targeted drug delivery[37, 38]. Due to the abundant expression of chemokines and chemokine receptors on stem cell membranes, they can interact with target cells, such as at the site of inflammation. Therefore, this type of nanocarrier can further enhance the potential for effective drug delivery in conditions such as inflammation[39].”
- When discussing the benefits of stem cells and their extracellular vesicles, it would be beneficial to compare them with other nanomaterials.
A: As suggested, we have added content of stem cells and their extracellular vesicles compared with other nanomaterials: “However, nanotechnology as a drug delivery approach still has limitations[33]. Magnetic nanoparticles, polymer nanogels, and other non-natural drug carriers are often categorized as "non-self" and are rapidly cleared by the immune system. They lack the ability to actively sense the disease environment, leading to insufficient accumulation of drugs in the desired organs or tissues, resulting in low targeting efficiency. Additionally, they present challenges such as immunogenicity and toxicity[34]. Therefore, there is a high demand for more precise nanomedical technologies to overcome these issues[35]. Stem cells, as natural carriers, can circumvent the issues associated with "non-natural" carriers.”
“As nanocarriers, EVs are endowed with cell-based biological structures and functions during the process of drug delivery, giving them inherent intercellular communication capabilities, enhanced cell-to-cell communication, and greater chemical stability. Moreover, EVs can achieve similar effects to synthetic nanocarriers like liposomes. Currently, widely studied drug carriers such as liposomes or polymer-based carriers are prone to phagocytosis by hepatic and splenic macrophages during in vivo circulation. Additionally, these carriers suffer from drawbacks such as short circulation time, poor stability, and low targeting specificity in the bloodstream[96]. The discovery of EVs ingeniously addresses these inevitable shortcomings of nanocarriers. Compared to synthetic nanocarriers, EVs can avoid being engulfed or degraded in circulation[97]. Being naturally secreted substances, they be able to overcome natural barriers such as the blood-brain barrier and inherently stable when circulating in the recipient[98-100], and their immunogenicity is also lower than other traditional carriers[101]. Stem cell-derived EVs exhibit inflammatory tropism, enabling them to exert therapeutic effects. In addition, EVs can achieve more precise drug targeting and delivery due to their complex and unique membrane structure[102]. Compared to synthetic nanocarriers, they can accommodate a greater variety of non-biological biomimetic materials, providing them with greater medical value. Therefore, as drug delivery vehicles, EVs possess natural advantages that make them safer, more stable, and more precise than other synthetic nanocarriers[103].”
- In Figure 3, the mentioned surface markers (e.g., CD9, CD81, and CD34) are transmembrane proteins with shorter or longer cytosolic domains. The current illustration gives the impression that they are anchored on the membrane surface and penetrate only one lipid layer. Please revise the illustration accordingly.
A: Thanks a lot for reviewers’ suggestion. We have revised the surface markers in Figure 3. Please refer to the revised manuscript.
- It would be helpful to include a list of abbreviations.
A: As suggested, we have listed a list of abbreviations at the end of the main text. Please refer to the revised manuscript.
Reviewer 2
- It is necessary for authors to list the subtitles and titles throughout the entire manuscript.
A: Thanks a lot for reviewers’ suggestion. we have listed the subtitles and titles in manuscript. Please refer to the revised manuscript.
- According to the abstract, the main idea of this review is to show the therapeutic potential of stem cells and extracellular vesicles (EVs) as nanocarriers. Sections 1 and 2 are fine. However, section 3 from line 93 to line 284, all these information is not related to the main objective of this review. Authors should remove it from the main text or shorten it, since the main part of the revision begins on Line 285. Bio-inspired nanocarriers by Stem cells-derived EVs. Here, authors must list the subtitles.
A: In addition to therapeutic potential of stem cells-derived EVs, we also summarized the application of biomimetic nanocarriers derived from stem cells in targeted drug delivery and discussed their advantages and challenges. Therefore, the main part in this manuscript begins from Section 3. Bio-inspired nanocarriers by stem cell. As suggested by reviewers, we have deleted some of the content in the third section and shortened it. And subtitles are also listed for Bio-inspired nanocarriers by Stem cells driven EVs. Please refer to the revised manuscript.
- Line 537-560. EVs modification. This section and Figure 3 should be immediately after Isolation of EVs (line 322-352). Change the order.
A: As suggested by reviewers, we have rearranged the positioning of this section and Figure 3. Please refer to the revised manuscript.
- Authors should include a table with the advantages and disadvantages of Stem cells-derived EVs and simulators of EVs (mimetic-nanovesicles, MNVs).
A: We have incorporated a table into the revised manuscript summarizing the advantages and disadvantages of Stem cells-derived EVs and simulators of EVs (mimetic-nanovesicles, MNVs). Please refer to the revised manuscript.
- Authors must include in the manuscript information about how this type of nanocarriers have been evaluated at a preclinical level in animal models, in what type of diseases they have been tested, and what type of drugs have been evaluated. Are there clinical trials currently underway? Include this information as well.
Also, it is recommended to include a table of these information.
A: According to reviewers’ comment, We have added the section 5. Application of stem cell-EVs in disease treatment: “5.1 Cardiovascular diseases Cardiovascular disease is a category of diseases that seriously affect human life and health, such as hypertension, coronary heart disease, mental infarction, etc. At present, the treatment of cardiovascular disease is mainly precision treatment. EVs as natural nano-carriers, could allow for the targeted delivery of therapeutic agents to the recipient cells, so as to achieve accurate drug administration. Zhao et al. discovered that EVs derived from MSCs can alleviate myocardial ischemia/reperfusion (I/R) injury in mice by modulating the polarization of M1 macrophages towards M2 macrophages. This crucial macrophage polarization is achieved through the involvement of miR-182, an important candidate mediator[182]. Other researchers have found that intracoronary injection of EVs derived from cardiosphere-derived cells (CDCs) carrying microRNA is safe and can improve cardiac function in a pig model of dilated cardiomyopathy (DCM)[183]. Additionally, in a mouse model of dilated cardiomyopathy, intravenous injection of exosomes derived from MSCs significantly reduces the number of pro-inflammatory macrophages in the blood and heart. This effect is attributed to the ability of exosomes to regulate macrophages and alleviate inflammation in the cardiac microenvironment[184]. Sun et al. demonstrated the role of EVs derived from MSCs overexpressing HIF-1α in promoting neovascularization and inhibiting fibrosis to maintain cardiac function in a rat model of myocardial infarction (MI)[185]. In addition, other studies have revealed that exosomes derived from mouse ESCs can enhance the survival of neovascularization and cardiomyocytes, reduce fibrosis after infarction, and restore cardiac function in a mouse model of acute myocardial infarction. The research indicates that this ability is associated with specific miR-294[186].” ... “As described above, extensive studies have demonstrated the potential clinical applications of EVs (Table 2). However, these approaches are still in the early stages of research and clinical application and there are still several challenges to be considered when transitioning from the laboratory to clinical practice. Standardization and improvement of the isolation and storage of EVs, as well as the development of quantitative, stable, and reproducible assays to assess their therapeutic potential, are essential to facilitate the translation of EVs from the laboratory to clinical applications.”Additionally, in the revised manuscript, we have summarized the aforementioned content and created a table that encompasses these information.
- Line 353. “Three types of EVs as nanocarriers”, change it by “Natural EVs”. List the subtitles.
A: As suggested by reviewers, we have made the change from “Three types of EVs as nanocarriers” to “Natural EVs” in the revised manuscript. And we have listed subtitles.
- Line 413. “However, it is noteworthy that microbubbles and apoptotic bodies…..”, is microbubbles or microvesicles?
A: We were really sorry for our careless mistakes. We have corrected it to “microvesicles” in the revised manuscript.
- Line 501. “MVNs as nanocarriers”, change it by “Simulators of EVs: mimetic-nanovesicles (MNVs). List the subtitle.
A: As suggested, we have made the change from “MVNs as nanocarriers” to “Natural EVs” in the revised manuscript. And we have listed subtitles.
Reviewer 3
The manuscript by Abudurexiti et al. deals with the issue of specifically targeted drug delivery, that recently became an interesting topic. So the concept of this review is really good, however, its realization is not. The Authors seem not to understand properly the term "bio-inspired" and "biomimetic", nevertheless they use these terms in many weird phrases. Some specific remarks are listed below.
Major remarks:
1/ the title is unclear and suggests that the review focuses on nanocarriers that are used to carry stem cells and EVs, please, rewrite it.
A: Thanks a lot for reviewers’ suggestion. The title has been rewritten as “Bio-Inspired nanocarriers derived from stem cells and their extracellular vesicles for targeted drug delivery”. Please refer to the revised manuscript.
2/ abstract and introduction sections are chaotic, they do not clearly state what is this manuscript about. Such terms as "natural biomimetic nanocarriers" (lines 17, 55, 56) or "bio-inspired stem cells" (line 22) are weird. Stem cells and EVs cannot be bio-inspired or bio-mimetic as they are, in fact, of biological origin.
A: According to the suggestion, we have revised the abstract to: “With their seemingly limitless capacity for self-improvement, stem cells have a wide range of potential uses in the medical field. Stem cell-secreted extracellular vesicles (EVs), as paracrine components of stem cells, are natural nanoscale particles that transport a variety of biological molecules and facilitate cell-to-cell communication, which have been also widely used for targeted drug delivery. These nanocarriers exhibit inherent advantages, such as strong cell or tissue targeting and low immunogenicity, which synthetic nanocarriers lack. However, despite the tremendous therapeutic potential of stem cells and EVs, their further clinical application is still limited by low yield and a lack of standardized isolation and purification protocols. In recent years, inspired by the concept of biomimetics, a new approach to biomimetic nanocarriers for drug delivery has been developed by combining nanotechnology and bioengineering. This article reviews the application of biomimetic nanocarriers derived from stem cells and their EVs in targeted drug delivery, and discusses their advantages and challenges in order to stimulate future research.”
Additionally, we have revised the introduction section as follows: “Stem cell membranes and EVs have been extensively studied and applied as nanocarriers for drug delivery. These cell-derived nanocarriers have excellent biocompatibility, good stability, low immunogenicity, and inherent targeting or tendency ability (e.g. penetration of biological barriers such as blood-brain barrier). Moreover, while maintaining the function of stem cells, they have long circulation time in vivo due to less phagocytosis, suggesting stem cell-derived EVs are outstanding candidates for targeted drug delivery and disease treatment[14]. However, their intrinsic drawbacks, including low extraction yields, poor recovery, purity and encapsulation efficiencies, restricted large-scale production of therapeutic exosomes in the clinic[15]. Therefore, bionic EV mimetic-nanovesicles (MNVs) have garnered attention. They contain key components of natural EVs and exhibit similar properties, with higher production yield compared to EVs[16]. In this review, we investigated the applications of biomimetic nanocarriers derived from stem cells and their EVs for targeted drug delivery, and discusses their advantages and challenges in order to stimulate future research.”
3/ paragraph numbering should be consequent in the whole manuscript
A: We have included paragraph numbering in the manuscript. Please refer to the revised manuscript.
Minor remarks:
1/ line 49 - shoul be "systemic administration"
A: As suggested, we have corrected it as follows: “Targeted drug delivery is a technique used to deliver drugs to specific tissues or organs of patients in systemic administration, while reduce accumulation in healthy tissues.”
2/ line 65 - "interact with all multicellular organisms" - what does it mean?
A: Thanks for reviewers’ suggestion. To avoid ambiguity, we have revise the sentence to: “Stem cells are undifferentiated cells with strong differentiation potential[17, 18].”
3/ line 80 - "nutritional factors" should be replaced with growth/trophic/biologicaly active factors
A: As suggested by reviewers, we have made the change from “nutritional factors” to “biologically active factors” in the revised manuscript. And we have listed subtitles.
4/ lines 201-202 - what are "cells secreted by mammals"?
A: Thanks for reviewers’ suggestion. To avoid ambiguity, we have removed this sentence.
5/ line 261 - what are "MSC extracted from iPSCs"?
A: We have referenced the description in the reference [73] “We have derived MSCs with an unlimited supply and uniform homing capacity to triple-negative breast cancer (TNBC) from human induced pluripotent stem cells (iPSCs).” (Zhao, Q., et al., Biomimetic nanovesicles made from iPS cell-derived mesenchymal stem cells for targeted therapy of triple-negative breast cancer. Nanomedicine: Nanotechnology, Biology and Medicine, 2020. 24: p. 102146). And as suggested by reviewers, we have revised it to “MSC derived from iPSCs” in the revised manuscript.
6/ lines 501 and further - please unify the abbreviation of MNVs
A: As suggested, the abbreviation of MNVs has been unified in the revised manuscript.
7/ line 604 - "the vesicle extracellular cells of stem cells" it is unclear, please correct
A: We were really sorry for our careless mistakes. We have corrected it to “the EVs of stem cells” in the revised manuscript.

Reviewer 2 Report
Comments are listed here for the author’s consideration to further improve the quality and overall impact of the manuscript.
1. It is necessary for authors to list the subtitles and titles throughout the entire manuscript.
2. According to the abstract, the main idea of this review is to show the therapeutic potential of stem cells and extracellular vesicles (EVs) as nanocarriers. Sections 1 and 2 are fine. However, section 3 from line 93 to line 284, all these information is not related to the main objective of this review. Authors should remove it from the main text or shorten it, since the main part of the revision begins on Line 285. Bio-inspired nanocarriers by Stem cells-derived EVs. Here, authors must list the subtitles.
3. Line 537-560. EVs modification. This section and Figure 3 should be immediately after Isolation of EVs (line 322-352). Change the order.
4. Authors should include a table with the advantages and disadvantages of Stem cells-derived EVs and simulators of EVs (mimetic-nanovesicles, MNVs).
5. Authors must include in the manuscript information about how this type of nanocarriers have been evaluated at a preclinical level in animal models, in what type of diseases they have been tested, and what type of drugs have been evaluated. Are there clinical trials currently underway? Include this information as well.
Also, it is recommended to include a table of these information.
6. Line 353. “Three types of EVs as nanocarriers”, change it by “Natural EVs”. List the subtitles.
7. Line 413. “However, it is noteworthy that microbubbles and apoptotic bodies…..”, is microbubbles or microvesicles?
8. Line 501. “MVNs as nanocarriers”, change it by “Simulators of EVs: mimetic-nanovesicles (MNVs). List the subtitle.
Minor editing of English language required.
Author Response

(The authors gave the same response as above.)

Reviewer 3 Report
The manuscript by Abudurexiti et al. deals with the issue of specifically targeted drug delivery, that recently became an interesting topic. So the concept of this review is really good, however, its realization is not. The Authors seem not to understand properly the term "bio-inspired" and "biomimetic", nevertheless they use these terms in many weird phrases. Some specific remarks are listed below.
Major remarks:
1/ the title is unclear and suggests that the review focuses on nanocarriers that are used to carry stem cells and EVs, please, rewrite it.
2/ abstract and introduction sections are chaotic, they do not clearly state what is this manuscript about. Such terms as "natural biomimetic nanocarriers" (lines 17, 55, 56) or "bio-inspired stem cells" (line 22) are weird. Stem cells and EVs cannot be bio-inspired or bio-mimetic as they are, in fact, of biological origin.
3/ paragraph numbering should be consequent in the whole manuscript
Minor remarks:
1/ line 49 - shoul be "systemic administration"
2/ line 65 - "interact with all multicellular organisms" - what does it mean?
3/ line 80 - "nutritional factors" should be replaced with growth/trophic/biologicaly active factors
4/ lines 201-202 - what are "cells secreted by mammals"?
5/ line 261 - what are "MSC extracted from iPSCs"?
6/ lines 501 and further - please unify the abbreviation of MNVs
7/ line 604 - "the vesicle extracellular cells of stem cells" it is unclear, please correct
English needs extensive revision
Author Response

(The authors gave the same response as above.)

Round 2
Reviewer 2 Report
There are some minor comments to improve the manuscript.
Line 738. “In addition, other studies have revealed that exosomes derived from mouse ESCs can enhance the survival of neovascularization…” …can enhance the survival of neovascularization? Correct paragraph.
Line 742. “The research indicates that this ability…” The research? Change by “This study…”
Section 5.4. Is there information regarding the use of these nanocarriers in the treatment of Covid 19 disease?
Indicate abbreviations in Table 2.
Line 866. “The secretome of these stem cells open up a new avenue for the deficiency of stem cell therapy.” This idea is not understood. Correct it.
Minor editing of English language required
Author Response
Dear Ms. Clarissa Chen
Thank you for your email dated 15 July 2023.
We have taken into consideration the constructive comments and suggestions of the reviewers for our manuscript, and have revised our manuscript accordingly. The corrections are highlighted in red in the revised manuscript.
Please find attached our revised manuscript, as well as our point-by-point response to the reviewers.
Again, thank you for taking the time to review this manuscript. We trust the present manuscript meets with the standards of Pharmaceutics.
Looking forward to hearing from you soon.
Yours sincerely, Zhi-xiang Yuan
We would like to express our sincere thanks to the editor and reviewers for the constructive and positive comments.
Response to the reviewers:
Reviewer 2
There are some minor comments to improve the manuscript.
Line 738. “In addition, other studies have revealed that exosomes derived from mouse ESCs can enhance the survival of neovascularization…” …can enhance the survival of neovascularization? Correct paragraph.
A: Thanks a lot for reviewers’ suggestion. According to the suggestion, we have revised this sentence to : “In addition, other studies have revealed that exosomes derived from mouse ESCs can enhance the neovascularization, promoted survival of cardiomyocytes, reduce fibrosis after infarction, and restore cardiac function in a mouse model of acute myocardial infarction.”
Line 742. “The research indicates that this ability…” The research? Change by “This study…”
A: As suggested by reviewers, we have corrected “The research” to “This study” in the revised manuscript.
Section 5.4. Is there information regarding the use of these nanocarriers in the treatment of Covid 19 disease?
A: According to reviewers’ comment, we have added the content: “Since the end of 2019, Coronavirus disease-19 (COVID-19) has become a new global threat that can lead to compromised immune systems, exacerbate the production of inflammatory cytokines, and cause coagulation disorders. According to reports, the miRNA carried by MSC-derived EVs (microvesicles and exosomes) can alleviate inflammatory factors, prevent tissue damage, and counteract coagulation disturbances. Therefore, MSC-EVs-miRNA has become a potential multi-target treatment approach for COVID-19[202].” And the related information has also been added in Table 2. Please refer to the revised manuscript.
Indicate abbreviations in Table 2.
A: As suggested, we listed the abbreviations below Table 2. Please refer to the revised manuscript.
Line 866. “The secretome of these stem cells open up a new avenue for the deficiency of stem cell therapy.” This idea is not understood. Correct it.
A: As suggested by reviewers, we have corrected it as follows:“The secretome of these stem cells has become a new hotspot and choice for disease treatment.”
Reviewer 3
The Authors have properly addressed all my remarks, and the corrected manuscript is considerably improved. The only mistake that was left uncorrected are lines 127-128: "stem cells interact with all multicellular organisms" - it has to be rewritten!
A: Thanks a lot for reviewers’ suggestion. We were really sorry for our careless mistake. To avoid ambiguity, we have revise the sentence to: “Stem cells are undifferentiated cells with strong differentiation potential[17, 18].”

Reviewer 3 Report
The Authors have properly addressed all my remarks, and the corrected manuscript is considerably improved. The only mistake that was left uncorrected are lines 127-128: "stem cells interact with all multicellular organisms" - it has to be rewritten!
only minor corrections are needed
Author Response

(The authors gave the same response as above.)
